# Predictive Modeling and Validation on Growth, Production of Asexual Spores and Ochratoxin A of *Aspergillus Ochraceus* Group under Abiotic Climatic Variables

**DOI:** 10.3390/microorganisms9061321

**Published:** 2021-06-17

**Authors:** Ahmed Abdel-Hadi, Bader Alshehri, Mohammed Waly, Mohammed Aboamer, Saeed Banawas, Mohammed Alaidarous, Manikandan Palanisamy, Mohamed Awad, Alaa Baazeem

**Affiliations:** 1Department of Medical Laboratory Sciences, College of Applied Medical Sciences, Majmaah University, Majmaah 11952, Saudi Arabia; b.alshehri@mu.edu.sa (B.A.); s.banawas@mu.edu.sa (S.B.); m.alaidarous@mu.edu.sa (M.A.); m.palanisamy@mu.edu.sa (M.P.); 2Department of Botany and Microbiology, Faculty of Science, Al-Azhar University, Assuit Branch, Assuit 71524, Egypt; m.fadl@tu.edu.sa; 3Department of Medical Equipment Technology, College of Applied Medical Sciences, Majmaah University, Majmaah 11952, Saudi Arabia; m.waly@mu.edu.sa (M.W.); m.aboamer@mu.edu.sa (M.A.); 4Department of Biomedical Sciences, Oregon State University, Corvallis, OR 97331, USA; 5Department of Biology, College of Science, Taif University, P.O. Box 11099, Taif 21944, Saudi Arabia; aabaazeem@tu.edu.sa

**Keywords:** *Aspergillus westerdijkiae*, *Aspergillus steynii*, water activity, temperature, growth rate, sporulation, ochratoxin A, polynomial regression

## Abstract

This study aimed to generate predictive models for growth, sporulation, and ochratoxin A (OTA) production under abiotic climatic variables, including temperatures (15–35 °C) and water activity levels (0.99–0.90 a_w_) by *Aspergillus ochraceus* group. The data were divided into three sets: one for training, one for testing, and the third one for model validation. Optimum growth occurred at 0.95 a_w_ and 25 °C and 0.95 a_w_ and 30 °C for *A. westerdijkiae* and *A. steynii*, respectively. Significantly improved *A. westerdijkiae* and *A. steynii* spore production occurred at 0.95 a_w_ and 20 °C and 0.90 a_w_ and 35 °C, respectively. *A. steynii* and *A. westerdijkiae* produced the majority of OTA at 35 °C and 0.95 a_w_ and 25–30 °C at 0.95–0.99 a_w_, respectively. The accuracy of the third-order polynomial regression model reached 96% in growth cases, 94.7% in sporulation cases, and 90.9% in OTA production cases; the regression coefficients (R2) ranged from 0.8819 to 0.9978 for the *Aspergillus ochraceus* group. A reliable agreement was reached between the predicted and observed growth, sporulation, and OTA production. The effects of abiotic climatic variables on growth, sporulation, and OTA production of *A. ochraceus* group have been effectively defined, and the models generated were responsible for adequately predicted and validated models against data from other strains within *A. ochraceus* group that had been published in the literature under the current treatments. These models could be successfully implemented to predict fungal growth and OTA contamination on food matrices for these strains under these conditions.

## 1. Introduction

Ochratoxin A (OTA) is one of the most important mycotoxins that can create a very wide range of health problems, such as nephrotoxic, immunotoxic, and teratogenic effects, in animals and humans (IARC 1993). Ochratoxin A can contaminate a wide range of foods, such as grains, cereal products, coffee, grapes, grape- products, dried fruits, nuts, cocoa, spices, and chocolate [1,2,3]. Ochratoxin A is predominantly produced by the genus Aspergillus. Two new ochratoxigenic species called *A. westerdijkiae* and *A. steynii* were separated from the *Aspergillus* section *Circumdati* that is more frequently described as the *A. ochraceus* species responsible for ochratoxins contamination in a wide variety of food products [4]. Consequently, since this differentiation has been demonstrated, there are several studies in which *A. westerdijkiae*, which is mainly responsible for OTA contamination, is isolated from a number of fresh materials, particularly coffee [5,6,7], grapes [8], paprika [9], barley [10], meat products [11]. It has been reported that *A. steynii* is the major OTA-producer within the section, with approximately 90% of the strains and toxins producing 1000 times more OTA than *A. ochraceus*, followed by *A. westerdijkiae* with 75% of its strains producing OTA and levels of production 100 times greater than *A. ochraceus* [12].

Previously, studies have investigated the ecology of species *A. ochraceus*. The single effect and interactions between water activity and temperature on growth, sporulation, OTA, and OTB of *A. ochraceus*, *A. steynii*, and *A. westerdijkiae* were investigated [13]. It has been shown that the growth of *A. steynii* and *A. westerdijkiae* was strongly affected by an increase in water activity and temperature, and neither growth nor sporulation resulted in good OTA production indicators for *A. steynii* or *A. westerdijkiae* [14]. Recent studies have been conducted with *Aspergillus* section *Circumdati* (*A. westerdijkiae*, *A. ochraceus*, *A. steynii*) and section *Nigri* species (*A. carbonarius*) in vitro and on stored coffee beans in relation to OTA production under climate change conditions [15,16]. Currently, a significant interest has been expressed in the impact of abiotic climatic variables (temperature, water availability, and CO_2_) and their interactions on the colonization of economically significant commodities with mycotoxic fungi and mycotoxins [17,18,19,20].

Mathematical models have been a powerful method for explaining microbial responses to diverse environmental conditions [21]. Predictive modeling of fungal species is still not given the same significance as bacterial growth models because of complexities in the estimation of fungal development and the collection of sufficient, appropriate, and reproducible data [22]. There are three types of microbial kinetic models: primary, secondary, and tertiary [23]. Primary models represent the changes in number of microorganisms in a population over time under certain conditions [24]. Secondary models explain the influence of environmental factors, such as water activity and temperature, on key parameters of the primary model. Tertiary models integrate primary and secondary models with a computer system to provide a comprehensive prediction tool [24]. In predictive microbiology, polynomial models have been successfully utilized to quantify the impacts of several environmental parameters on fungal growth, most notably a_w_, temperature, and pH. They are commonly constructed using multiple linear regression analyses and allow for the assessment of environmental parameters and their interactions [22]. A significant amount of information on models is now available to predict fungal growth and development of mycotoxins in food matrices [25,26,27]. The lack of validity of the tested models was a disadvantage for the developed models. Practically, no reliable data were available for developing models to predict and validate sporulation and OTA production in the *A. ochraceus* group. Thus, the aims of this work were (a) to generate predictive models for the combined effect of temperature and water activity (a_w_) on growth, spore production, and OTA production of *A. westerdijkiae* and *A. steynii* and (b) to validate the generated models against published data from other strains within the same genus in the literature.

## 2. Materials and Methods

### 2.1. Fungal Strains

Two ochratoxigenic strains very similar in morphology (colony and microscopic characteristics), but with variations in their responses to environmental conditions in terms of OTA biosynthesis regulation [28] from the group of *A. ochraceus* within section *Circumdati* (*A. steynii* IBT 23096, and *A. westerdijkiae* IBT 23971) were used in the experimental work of this study. Prof. J. Frisvad (Tech. Univ. of Denmark, Lyngby, Denmark) kindly supplied the strains [4].

### 2.2. Media Preparations

The water activity levels of yeast extract sucrose (YES) medium, which is a toxin-induced medium (20.0 g L^−1^ yeast extract, 150.0 g L^−1^ sucrose, 15 g L^−1^ agar), was modified with glycerol to three levels (0.99, 0.95, and 0.90 a_w_). Known quantities of glycerol/water solutions (51 g L^−1^ in case of 0.99 a_w_, 231.5 g L^−1^ in 0.95 aw, and 378.9 g L^−1^ 0.90 a_w_) at these aw levels were prepared and applied to the medium [29]. Media were autoclaved at 121 °C for 20 min. The same treatments were placed in separate polyethylene bags and confirmed using AQUALAB ^®^ 3TE, USA, for verification of actual values.

### 2.3. Culturing and Growth Rate Assessment

Petri plates with modified media were inoculated with 5 μL of 10^6^ spore suspension of both strains and incubated at 15, 20, 25, 30, and 35 °C. Three replicates were conducted for all studies. Those at the same level of water activity were placed in sealed polyethylene bags. The diameters of the growing colony of replicated plates were measured in two directions at right angles with each other. Colony growth measurements were recorded daily for 10 days until the colonization of the Petri plates was completed in some treatments [30]. The same plates used for growth measurements were used for OTA determinations. The growth rate was determined by plotting the radial mycelium growth across time and using the slope of the linear growth process to obtain the radial growth rate (mm/day) [31].

### 2.4. Measurements of Asexual Spores

The species were inoculated on YES medium covered with cellophane (8.5 cm, P400, Cannings Ltd., Bristol, UK), allowing the whole mycelium of the colony to be separated. The entire mycelium was suspended in sterile water (10 mL) containing a wetting agent (Tween 80, 0.1%) to wet the spores after 10 days of incubation. Three replicates were conducted for all treatments. Spores were filtered through sterile glass wool, and the filtrate was centrifuged to produce a spore pellet. The number of spores per cm of the colony at the end of the incubation time was determined using a hemocytometer and a binocular Olympus Microscope connected with a camera and LCD screen [13].

### 2.5. Estimation of Ochratoxin A

Three plugs (3 mm in diameter) were taken out from the agar of each replicate across the colony surface, with one in the center and two on both sides using a cork borer. The plugs were put into 2 mL Eppendorf tubes and weighed. OTA was extracted by adding 1 mL of HPLC grade methanol and shaking for 1 hour. The mycelial residue was removed using centrifugation. The extracts were directly derived into amber HPLC vials. Samples were collected and analyzed using HPLC with a 470-fluorescence detector (Agilent, USA) (λexc 333 nm, λem 460 nm) and a C 18 column (Luna Spherisorb ODS2 150 × 4.6 mm, 5 µm) operated using Waters Millennium 32 software. The analysis was performed at a flow rate 1 mL min^−1^ with a mobile phase of 41% water: 57% acetonitrile: 2% acetic acid with a run time of 15 min with a detection limit of 0.5 ng g^−1^ and quantification limit of 2.4 ng g^−1^ [13,18].

### 2.6. Modeling of Fungal Growth, Sporulation, and Ochratoxin A

The data were divided into three sets. The data produced from the experimental work with *Aspergillus westerdijkiae* IBT 23971 were used for the training set, the data produced from the experimental work with *Aspergillus steynii* IBT 23096 were used for the testing and derived data from *Aspergillus ochraceus* IBT 11952, *Aspergillus steynii* IBT 22339, and *Aspergillus westerdijkiae* IBT 10738 [13] were used to validate the models. To generate a model capable of predicting fungal growth rate, sporulation, and OTA production as a function of a_w_ and temperature, the following processes were applied using Microsoft Excel software and MATLAB2018a [32,33].

#### 2.6.1. Normalization

The growth, sporulation, and OTA production data from *Aspergillus westerdijkia* at a_w_ (0.95) and temperatures (20, 25, 30, 35 °C) were normalized with the general expression:(1)$$X^{\prime }=\frac{X-X_{min}}{X_{max}-X_{min}}$$
where *X_min_* is the minimum value of input variables (growth, sporulation, and ochratoxin A), *X_max_* is the maximum value of the input variables, and $X^{\prime }$ is the normalized calculated value.

#### 2.6.2. Linear Interpolation

The data of growth, sporulation, and OTA production of *A. westerdijkiae* were interpolated using the linear interpolation formula
(2)$$y=y_{0}+\left(x-x_{0}\right)\frac{y_{1}-y_{0}}{x_{1}-x_{0}}$$

In the equation *y* is the variables data (growth, sporulation, and OTA), *y*_0_ and *y*_1_ are the two closest approximations to *y*, *x* is temperature data, and *x*_0_ and *x*_1_ are the two closest approximations to *x*.

#### 2.6.3. Polynomial Regression

A third-order polynomial response surface model (Equation (3)) was fitted to the dependent variables (growth rate, sporulation, and OTA) with the independent variables (temperature), at a fixed a_w_ (0.95):(3)$$y=a_{0}+a_{1}x+a_{2}x^{2}+a_{3}x^{3}+\cdots .+\varepsilon$$

In the equation *a*_0_…*a* represent the estimated regression coefficients, *ε* is an unobserved random error with mean zero conditioned on a scalar variable *x* (temperature), and *y* = λ or μ. In all-purpose, the expected value of *y* as an nth degree polynomial, yielding the general polynomial regression model. Polynomial regression was used to study the behavior for each type of the *Aspergillus ochraceus* group.

#### 2.6.4. Nonlinear Regression

The effect of temperature on the growth rate, sporulation, and OTA production was predicted using the nonlinear regression equation:(4)ln(y)=ln(a)+bx+u

In the equation, *a* and *b* are the parameters, and *u* is the multiplicative error, suggesting the estimation of the unknown parameters by linear regression of ln(*y*) on *x*.

### 2.7. Model Testing and Validation

The developed models from *A. westerdijkiae* IBT 23971 data were tested on *A. steynii* IBT 23096 data and validated against external data from Abdel-Hadi and Magan [13] for *Aspergillus westerdijkiae* 10738, *Aspergillus steynii* 22339, and *Aspergillus ochraceus* 11952 using the same temperatures employed for the model and a_w_ level (0.95). Performances of the generated models were evaluated using the mean squared error (MSE) and the root mean squared error (RMSE), as suggested by Ross [34], where the RMSE value near zero indicates a better model fit:(5)$$\mathrm{RMSE}=\sqrt{\frac{1}{N}\;\sum_{i=1}^{N}{\left(M_{i}-A_{i}\right)}^{2}}$$
where: ***RMSE*** = root mean square error, ***N*** = sample size, ***M_i_*** = predicted values, and ***A_i_*** = observed values.

### 2.8. Statistical Analysis

All experiments for growth, sporulation, and ochratoxin A production were conducted and repeated with three replicates for measurements of asexual spores and three replicates for growth measurements and OTA estimations. Statistical analyses were performed using Statistica, version 8 for two-way ANOVA, followed by Fisher’s LSD test to compare the means for all treatments that were significantly different at *p* < 0.05.

## 3. Results and Discussion

### 3.1. Growth Assessment in Relation to Abiotic Climatic Variables

Figure 1 shows growth rate results for *A. steynii* 23096 and *A. westerdijkiae* 23971 at varying a_w_ and temperature levels. The optimum growth for both species was around 0.95 a_w_, but it occurred at different temperatures. Optimal growth occurred at 25 °C for *A. westerdijkiae* 23971, and at 30 °C for *A. steynii* 23096. Generally, both species exhibited similar growth rates at 0.90 a_w_. *Aspergillus steynii* 22339 grew faster than *A. westerdijkiae* 23971 at higher water activity (0.95 and 0.99). There was no growth for both strains at 15 °C with 0.90 and 0.95 a_w_. Statistical analysis of the data-sets demonstrated that single and interacting factors were all significant for a_w_, temperature treatments, and their interactions (Table 1). A previous study showed that *A. steynii* and *A. westerdijkiae* growth rates were significantly reduced at lower a_w_ values (0.91 and 0.93) and reached the optimum at a_w_ = 0.97 when they grew in medium green coffee [5]. A further study examined the effect of ecophysiological factors on growth, sporulation, and OTA development of *A. steynii* and *A. westerdijkiae* in CYA medium (Czapek Yeast Agar) and food products media. The authors found similar permissive and optimum growth conditions for *A. steynii* and *A. westerdijkiae,* where the growth increased with temperature in both species, and fungal growth was significantly reduced at the lowest a_w_ [14]. Similarly, the optimum growth of *A. westerdijkiae* was usually at a_w_ rates between 0.93 and 0.97, and it was significantly reduced at all temperatures measured in medium dry-cured ham with lower a_w_ [11]. Another study examined the influence of temperature and a_w_ on *A. westerdijkiae* growth in stored barley, and the study suggested that a_w_ is a critical factor for fungal growth in this product [35]. The European Commission suggests that changes in Mediterranean zones may shift to a temperature increase of 4–5 °C associated with prolonged dry seasons [36]. High temperatures reaching 30 °C and water stress conditions (0.95 a_w_) affected *A. steynii* and *A. westerdijkiae* growth positively, while low temperature (20 °C or less) and high drought (0.90 a_w_), negatively. Regarding this, Akbar et al. [16] suggested that lag phases prior to growth and growth rates of *A. westerdijkiae*, *Aspergillus steynii* grown on coffee-based media and stored coffee beans were significantly influenced by climate change (CC) factors, and these strains tended to become more tolerant of elevated temperature. A recent study by Akbar et al. [18] reported that *A. westerdijkiae* strains could colonize rapidly on drying green coffee in a range of 25–35 °C with reduced growth at 0.90 a_w_.

### 3.2. Sporulation Assessment in Relation to Abiotic Climatic Variables

There were substantial differences between *A. steynii* and *A. westerdijkiae* at all the temperatures tested, with no spores produced at 0.99 a_w_. Spore production was significantly enhanced in *A. westerdijkiae* 23971 at 0.95 a_w_ and 20 °C, and *A. steynii* 23096 had remarkably increased sporulation ability at 0.90 a_w_ and 35 °C (Figure 2). Water activity had a significant effect on sporulation in both species, where they displayed dramatic growth responses, with slower mycelium growth and higher spore production when water stress was imposed but at different temperatures. The effect of single and two-way interactions is shown in Table 1. There were statistically significant differences due to the single, interacting factors of a_w_ and temperature in species. Many studies have been reported in the literature regarding asexual spore production of various mycotoxigenic fungi, such as *A. flavus*, *A. ochraceus* group, *A. niger,* and *Fusarium verticillioides,* on synthetic media and food matrix [37,38,39]. Results similar to our study were obtained by one research group [11] with *Penicillium nordicum* and *A. westerdijkiae* sporulation grown on a dry-cured ham-based medium. The authors reported that the highest number of spores for *P. nordicum* and *A. westerdijkiae* Type-strain spores were produced at 0.97 aw and 20 °C, and 0.90 a_w_ and 30 °C, respectively. There was a difference in the number of spores produced at 0.90 a_w_ and 30 °C compared to our study. This may because the incubation time was longer. In contrast, other studies reported that *A. steynii*, and *A. westerdijkiae* on food product-based media and *A. flavus* grown on maize stalks substrates showed abundant sporulation at 0.99 a_w_ [16,40]. This may explain the effect of substrate or strain type on spore production. Zhang et al. [41] studied the activity of *A. flavus* and the transcriptome at 0.99 and 0.93 a_w_ and 28 °C grown on a conducive YES medium. They reported that both conidiation and AFB1 biosynthesis decreased at 0.93 a_w_ when compared to 0.99 a_w_ due to the upregulation of 3156 genes and downregulation of 2206 genes identified between 0.99 and 0.93 a_w_ treatments. They suggested that *A. flavus* underwent an extensive transcriptome response during the variation in water activity levels.

### 3.3. Ochratoxin A Assessment in Relation to Abiotic Climatic Variables

Figure 3 shows OTA production results of *A. westerdijkiae* 23971 and *A. steynii* 23096 under different a_w_ and temperature regimes on YES medium. Interestingly, *A. westerdijkiae* 23971 produced greater amounts of OTA than *A. steynii* 23096 at all examined temperatures and a_w_ levels. In contrast to this, Gil-Serna et al. [12] reported that *A. steynii* had the capacity to produce OTA at higher levels than *A. westerdijkiae*. This could be due to genetic diversity and strain specificity and shifts our opinion that *A. steynii* produces more OTA than A. *westerdijkiae*. In addition, the increased levels of OTA observed in *A. westerdijkiae* 23971 could be as YES is a more conducive medium for toxin production for *A. westerdijkiae* 23971 but not for *A. steynii* 23096. *A. steynii* 23096 and *A. westerdijkiae* 23971 produced the highest amounts of OTA at 35 °C and 0.95 a_w_ and at 25–30 °C and 0.95–0.99 a_w_, respectively. Our results are similar to those of another group which reported that *A. westerdijkiae* and *A. steynii* produced most of the OTA at temperatures above 24 °C and 0.96–0.99 a_w_ when grown on food product-based media [16]. In our study, there was a close association between fungal growth and OTA production. Therefore, fungal growth could be used as an indicator of OTA contamination by these strains. In contrast to this, another group reported that there was no significant correlation between fungal growth and OTA production of *A. westerdijkiae* and *A. steynii* grown on food product-based media [16]. Similarly, Gil-Serna et al. [14] stated that neither growth nor sporulation resulted in good OTA production indicators for *A. steynii* or *A. westerdijkiae* grown on food matrices-based media. This may be because food-based medium with different carbon sources had a critical impact on the OTA production ability of Aspergillus species. Available information suggests that slightly elevated temperature and interaction with water stress may stimulate OTA produced by fungi and allow new OTA fungal strains to appear in food matrices and become more dominant [15]. Akbar et al. [16] examined the effect of CC factors and their interactions on OTA production in vitro and on stored coffee beans by *A. wetserdijkiae*, *A. steynii*, *A. ochraceus*, and *A. carbonarius*. They suggested that OTA production stimulated by *A. westerdijkiae* (35 °C) and *A. niger* (30 °C) while reduced by *A. carbonarius* in stored coffee. This shows the differential ability of OTA producers to adapt to CC factors. A recent study by Lappa et al. [42] examined the effect of temperature and water stress on the expression of genes required for OTA production in *A. carbonarius*. They showed that, while water activity was the critical factor influencing OTA production, temperature was the only triggering factor to influence the transcription of OTA biosynthesis gene. In addition, an alternative approach is to precede models of anticipated ochratoxins contamination. Cervini et al. [43] reported that the growth of *A. carbonarius* isolated from grapes was relatively unaffected on grape-based media by interacting abiotic CC-related factors (temperature, water availability, and CO_2_), while ochratoxin production was stimulated due to the increase in ochratoxin relative gene expressions. Although several studies have examined the effects of different abiotic climatic factors and their interactions on *Aspergillus westerdijkiae*, *Aspergillus ochraceus*, *Aspergillus steynii*, and *Aspergillus niger* species/strains [13,18,37], no data on toxin regulation at a molecular level and model prediction in vitro and on food matrices are available to date, and this knowledge gap needs to be addressed [44].

### 3.4. Predictive Modeling and Validation

Since *A. westerdijkiae* IBT 23971 produced more OTA under all tested conditions and can produce asexual spores under 0.95 a_w_ at all temperatures, the data for its growth, sporulation, and OTA production under 0.95 aw was selected to generate the model. Sporulation of *A. westerdijkiae* was possible only at 0.95 a_w_. Therefore, no data modeling was performed at 0.99 and 0.90 a_w_ because models, with the exception of no-data, generate a good fit for experimental data [45]. Initially, the colony growth rate, sporulation, and OTA production were modeled and normalized vs. temperature with a normalization formula (Equation (1)). The experimentally observed data of growth rate, sporulation, and OTA production were fitted with rescaled normalized data (Figure 4). A linear interpolation (Equation (2)) method was used to estimate growth rate, sporulation, and OTA production under estimated temperatures (Figure 5). A third-order polynomial equation (3) was used to describe the effect of temperature on the growth rate, sporulation, and OTA production (Figure 6). There was a strong positive correlation between growth rate, sporulation, and OTA production of *A. westerdijkiae* IBT 23971 and temperature, with regression coefficients (*R*^2^) of 0.96, 0.88, and 0.977, for growth rate, sporulation, and OTA production, respectively. The third-polynomial regression model was applied for all the strains where *R*^2^ ranged from 0.9619 to 0.9797 for growth rate, 0.8819 to 0.9903 for sporulation, and 0.977 to 0.9978 for OTA production (Table 2).

To our knowledge, this is the first study to use third-order polynomial regression models for the prediction of growth, sporulation, and OTA production under different temperatures and 0.95 a_w_ using *Aspergillus ochraceus* group data. This study has provided a useful model that can be applied to OTA producers in vitro and validated on stored food that can be contaminated with OTA producers. Garcia et al. [22] reported that the polynomial model is the most common model used to describe fungal growth because of its flexibility and simplicity. They found good prediction ability given by high *R*^2^ and low RMSE. *Aspergillus westerdijkiae* IBT 23971 growth, sporulation, and OTA production showed significant differences due to the single and interacting factors of a_w_ and temperature (Table 1). Detailed information about sporulation of the *Aspergillus ochraceus* group is very important to understand fungal colonization. The present study suggests that the change in climate factors can alter the sporulation of these strains in an unpredictable way. The utilization of sporulation data under CC factors will be a significant addition for future studies in developing successful models to understand the fungal contamination in foodstuffs. Previous studies reported that optimal water activity for growth, sporulation, and OTA production for OTA producers is 0.95 a_w_ for culture media and food matrices [37,39].

The equations derived from the third-order polynomial regression model of *A. westerdijkiae* IBT 23971 were tested against *A. steynii* IBT 23096 data. The accuracy of the generated model was 96% for growth rate, 94.7% for sporulation, and 90.9% with OTA production with RMSE 0.039, 0.054, and 0.09, respectively. The model was then validated against external data from Abdel-Hadi and Magan [13] for *Aspergillus westerdijkiae* 10738, *Aspergillus steynii* 22339, and *Aspergillus ochraceus* 11952 over a wide range of temperature and 0.95 a_w_. RMSE values for growth rate ranged from 0.0390 to 0.2296, sporulation was 0.0539 to 0.5562, and OTA was 0.0912 to 0.4332 for all tested strains (Table 3). RMSE is an indicator of residual differences between the predicted and the observed data [46]. The RMSE of an acceptable developed model for *A. flavus* growth reportedly ranged from 0.1 to 0.849 [47]. We used an effective modeling approach to examine the effects of abiotic climatic variables on growth, sporulation, and OTA production, where RMSE values for this study ranged from 0.0390 to 0.5562. Another study implemented a second-order polynomial model with OTA aggregation as a variable of time, a_w_, temperature, and carbendazim concentrations. The model recorded a reasonable coefficient of similarity between observed and expected model values, excluding higher and lower OTA amounts where predictability was harder [48]. Other studies reported that models which applied culture media performed poorly when compared to food matrices [49]. This may be due to deficiencies in the model or other factors that influence fungal responses.

The accuracies of the derived models for the three validated strains extracted from literature (*A. westerdijkiae* 10738, *A. steynii* 22339, and *A. ochraceus* 11952) reached 92.9% in growth cases, 71.7% in sporulation cases, and 78.8% in OTA production cases. All the strains showed a good similarity for both observed and predicted growth, sporulation, and OTA production at different temperatures. To show the similarity of both observed and predicted results, data of *Aspergillus westerdijkiae* 10738 were plotted as shown in Figure 7, while data for the other strains were not presented. In all the conditions examined, with temperature changes, a reasonable agreement was reached between the predicted and observed growth, sporulation, and OTA production with little downshift of the predicted growth and OTA production at 30 °C and 25 °C for sporulation. Paola et al. [26] created and assessed a computational model for functional simulation of the *Aspergillus carbonarius* life cycle in grapes during the growing season, along with OTA production in berries. They reported that model validation was not achievable due to poor OTA contamination results. However, there were significant differences in the model output index between lower and higher risk areas.

One of the most critical elements of model generation is to ensure that the model’s predictions are adapted to actual scenarios. Therefore, model validation is important. It requires a comparison between model predictions and the actual observations, which should be different from the data used to create the original model [50]. External validation of existing models to describe fungal growth and mycotoxin production are scarce. Marín et al. [51] validated models for probability growth and OTA production by *A. carbonarius* related to moisture content and temperature on pistachio nuts. The authors found that the accuracy of the model was shown to predict 73–91% of growth cases, while the probability of the presence of OTA was accurately predicted in 90% of cases. Our model showed good performance when applied to three species of the *Aspergillus ochraceus* group on culture media. These models need to be validated on food matrices that can be contaminated with OTA producers.

## 4. Conclusions

Our findings showed that the growth rate, sporulation, and OTA production of the *A. ochraceus* group as a function of abiotic climatic variables (temperature and water activity) could be predicted quantitatively in other or similar environments using the developed models. Moreover, the third-order polynomial regression model approached reasonable agreement between the predicted and observed growth rates, sporulation, and OTA production, where *R*^2^ ranged from 0.8819 to 0.9978. The optimal abiotic climatic variables for growth rate, sporulation, and OTA production closely resemble previous studies. However, it is necessary to validate the models in food matrices, especially for new strains to evaluate the consequences of lapses in the process and storage conditions and estimate the degree to which abiotic climatic variables and their interactions influence storage stability and food safety without the need for a long-term storage study. By controlling abiotic climatic variables at different places in the storage unit and integrating these data into the models, it is possible to estimate if these conditions are favorable to the growth of *A. ochraceus* group or OTA production. Additionally, predictive models for other factors, such as CO_2_ level, use of fungicides, and genomic information that may have a role in OTA management, need to be investigated.

## Figures and Tables

**Figure 1 microorganisms-09-01321-f001:**
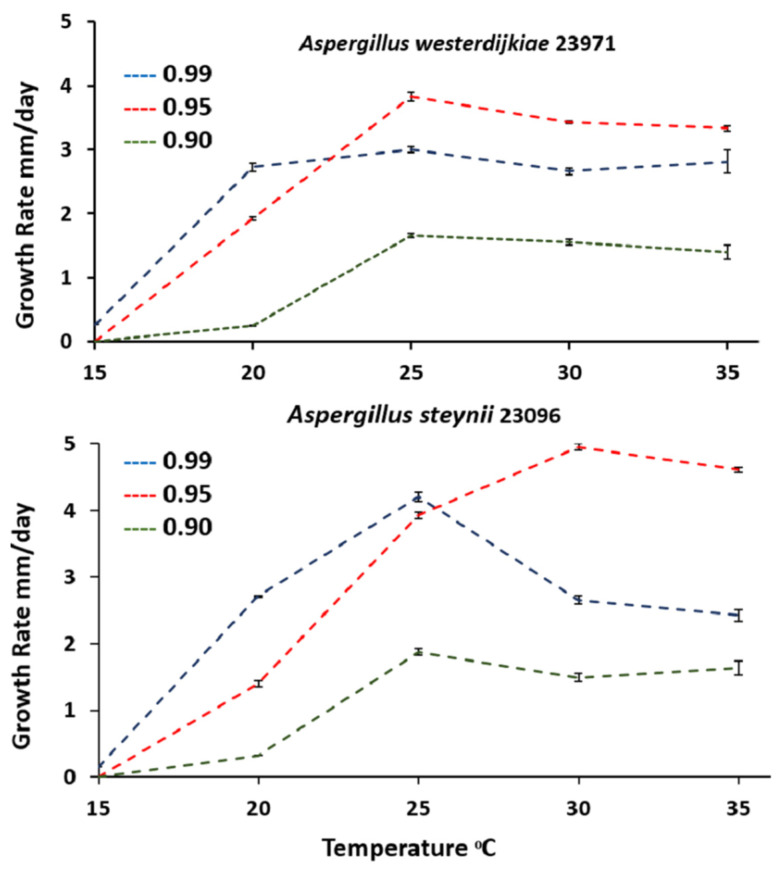
*A. westerdijkiae* IBT 23971 and *A. steynii* IBT 23096 growth rate in relation to temperature and water activity (a_w_) in the YES medium. Vertical bars show standard errors for all strains.

**Figure 2 microorganisms-09-01321-f002:**
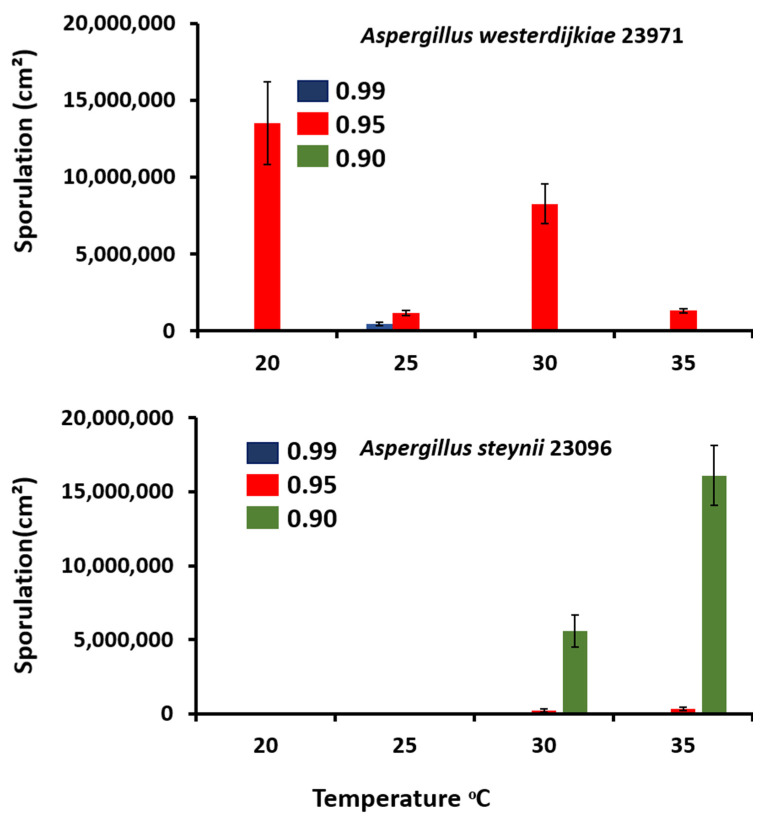
Sporulation of *A. westerdijkiae* IBT 23971 and *A. steynii* IBT 23096 in relation to temperature and water activity (a_w_) in the YES medium. Vertical bars show standard errors for all strains.

**Figure 3 microorganisms-09-01321-f003:**
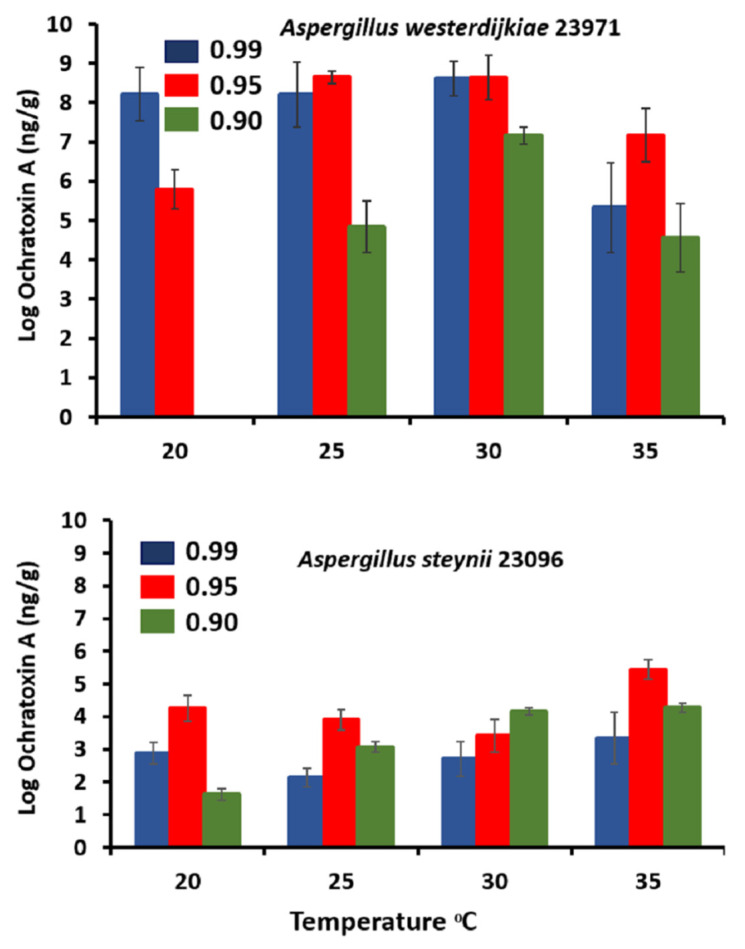
Ochratoxin A of *A. westerdijkiae* IBT 23971 and *A. steynii* IBT 23096 in relation to temperature and water activity (a_w_) in the YES medium. Vertical bars show standard errors for all strains.

**Figure 4 microorganisms-09-01321-f004:**
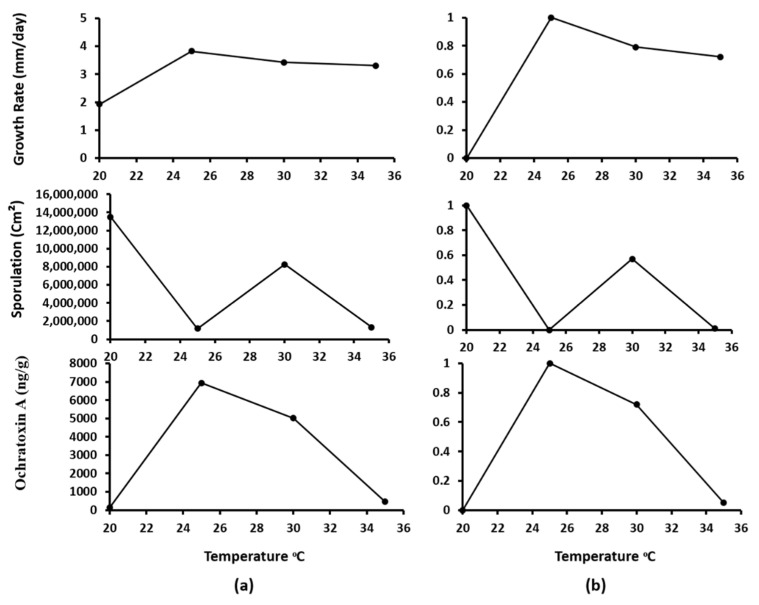
Growth rate, sporulation, and ochratoxin A data of *A. westerdijkiae* IBT 23971 before (**a**) and after (**b**) normalization in relation to temperature at 0.95 a_w_ on YES medium.

**Figure 5 microorganisms-09-01321-f005:**
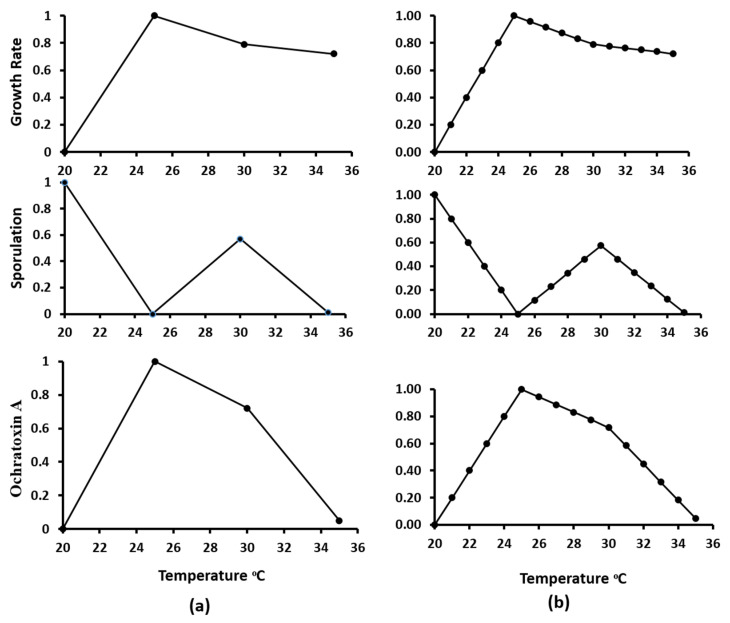
Growth rate, sporulation, and ochratoxin A data of *A. westerdijkiae* IBT 23971 before (**a**) and after (**b**) linear interpolation in relation to temperature at 0.95 a_w_ on YES medium.

**Figure 6 microorganisms-09-01321-f006:**
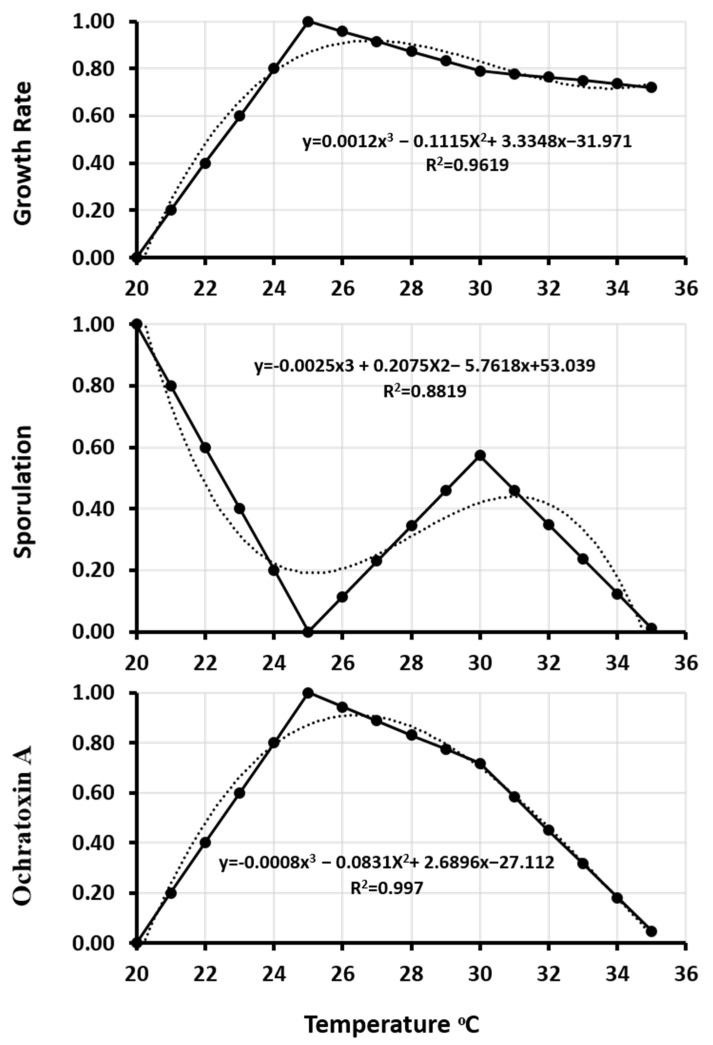
Multiple degree polynomial regression of growth rate, sporulation, and ochratoxin A of *A. westerdijkiae* IBT 23971 in relation to temperature at 0.95 a_w_ on YES medium.

**Figure 7 microorganisms-09-01321-f007:**
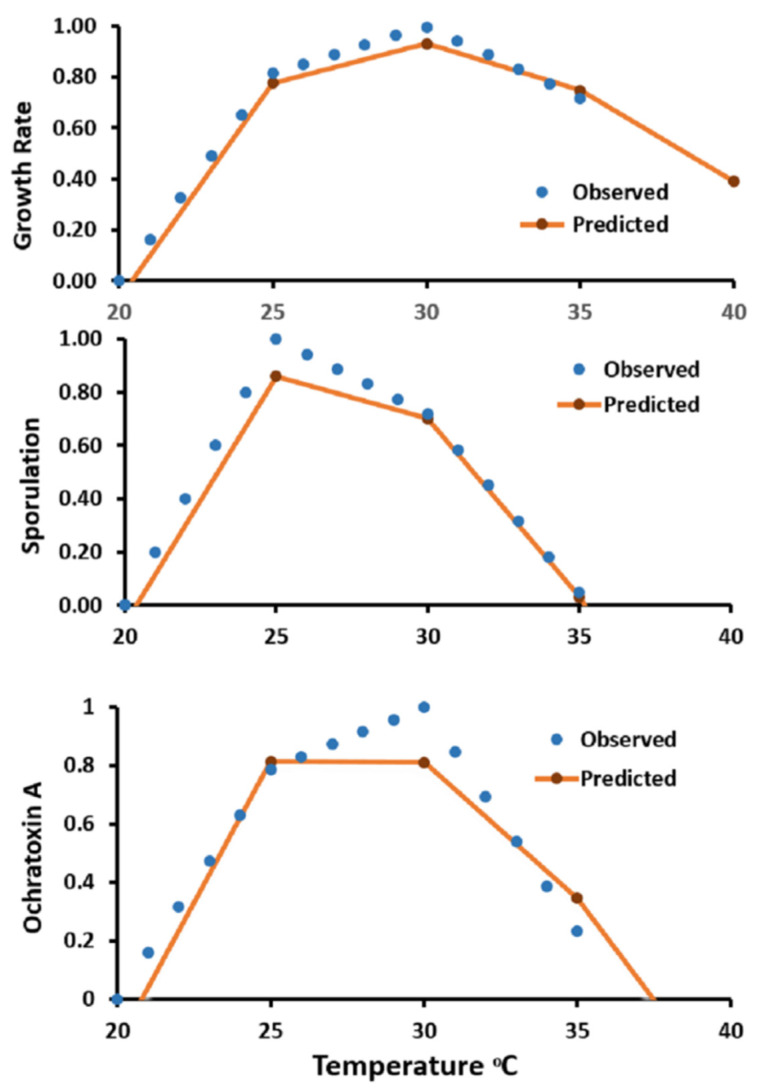
Comparison between observed and predicted values of growth, sporulation, and ochratoxin A of *A. westerdijkiae* IBT 10738 in relation to temperature at 0.95 a_w_ on a YES medium.

**Table 1 microorganisms-09-01321-t001:** Variance analysis of the influence of temperature, a_w_ and their interaction on growth rate, spore production, and ochratoxin A production of *A. westerdijkiae* and *A. steynii*.

		*Aspergillus Westerdijkiae* 23971				*Aspergillus Steynii* 23096			
	**Factor**	**DF**	**MS**	**F**	**P**	**DF**	**MS**	**F**	**P**
Growth rate	Temperature	4	11.2329	2693.48	0.00	4	17.0717	6257.19	0.00
	a_w_	2	10.2443	2456.42	0.00	2	14.5386	5328.76	0.00
	Temperature × a_w_	8	0.9489	225.61	0.00	8	2.7519	1008.65	0.00
Sporulation	Temperature	3	3.420 × 10^13^	3.0773	0.046	3	5.988 × 10^13^	137.093	0.00
	a_w_	2	1.438 × 10^14^	12.9402	0.00	2	1.145 × 10^14^	262.299	0.00
	Temperature × a_w_	6	3.632 × 10^13^	3.26787	0.017	6	5.654 × 10^13^	129.375	0.00
Ochratoxin A	Temperature	3	108.844	368.25	0.00	3	96.501	251.404	0.00
	a_w_	2	66.608	225.35	0.00	2	70.678	182.566	0.00
	Temperature × a_w_	6	24.262	82.69	0.00	6	8.064	21.008	0.00

Where DF = Degree of freedom; MS = Mean Square; F = Frequency; P = Probability.

**Table 2 microorganisms-09-01321-t002:** Predictive model equations based on third-order polynomial regression.

Strains		Equation	*R* ^2^
*Aspergillus westerdijkiae* 23971	Growth rate	y = 0.0012x^3^ − 0.1115x^2^ + 3.3348x − 31.971	0.9619
	Sporulation	y = −0.0025x^3^ + 0.2075x^2^ − 5.7618x + 53.039	0.8819
	Ochratoxin A	y = 0.0008x^3^ − 0.0831x^2^ + 2.6896x − 27.112	0.977
*Aspergillus westerdijkiae* 10738	Growth rate	y = 0.0001x^3^ − 0.0227x^2^ + 0.954x − 11.222	0.9927
	Sporulation	y = −0.0002x^3^ + 0.0042x^2^ + 0.3017x + 5.9932	0.9871
	Ochratoxin A	y = −0.0004x^3^ + 0.0153x^2^ + 0.0258x − 3.7228	0.9848
*Aspergillus steynii* 22339	Growth rate	y = 0.001x^3^ − 0.0904x^2^ + 2.673x −25.514	0.9797
	Sporulation	y = −0.0009x^3^ + 0.0654x^2^ +1.5539x + 112.197	0.984
	Ochratoxin A	y = 0.0008x^3^ − 0.0687x^2^ + 2.09792x − 20.24	0.9883
*Aspergillus steynii* 23096	Growth rate	y = 0.00003x^3^ − 0.0106x^2^ + 0.5627x − 7.321	0.9967
	Sporulation	y = −0.001x^3^ + 0.0849x^2^ − 2.2961x + 19.965	0.9903
	Ochratoxin A	y = 0.0004x^3^ − 0.0302x^2^ + 0.8883x − 8.5945	0.9978
*Aspergillus ochraceus* 11952	Growth rate	y = 0.0011x^3^ − 0.0934x^2^ + 2.6015x − 23.685	0.9734
	Sporulation	y = 0.0013x^3^ − 0.0997x^2^ − 2.4917x + 20.127	0.9805
	Ochratoxin A	y = 0.0009x^3^ − 0.062x^2^ + 1.4539x − 11.226	0.9844

Where *R*^2^ = Regression coefficients.

**Table 3 microorganisms-09-01321-t003:** Model validation of *Aspergillus ochraceus* group.

Strain	Growth Rate		Sporulation		Ochratoxin A	
	**MSE**	**RMSE**	**MSE**	**RMSE**	**MSE**	**RMSE**
*Aspergillus steynii* 23096	0.0015	0.039	0.0029	0.0539	0.0083	0.0912
*Aspergillus westerdijkia* 23971	0.0273	0.1654	0.3094	0.5562	0.1877	0.4332
*Aspergillus ochraceus* 11952	0.0527	0.2296	0.056	0.2367	0.1516	0.3894
*Aspergillus steynii* 22339	0.0058	0.0764	0.0832	0.2885	0.045	0.2123
*Aspergillus westerdijkia* 10738	0.0087	0.0934	0.2217	0.4708	0.1138	0.3373

## Data Availability

The data used to support the findings of this study are available from the corresponding author upon request.

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
