# Peer review of "Predictive Modeling and Validation on Growth, Production of Asexual Spores and Ochratoxin A of Aspergillus Ochraceus Group under Abiotic Climatic Variables"

_microorganisms, 2021, doi:10.3390/microorganisms9061321_

Round 1
Reviewer 1 Report
microorganisms-1207173-peer-review-v1
Reviewer summary.
The goal of this work is to develop predictive models that can be used to understand the effect of water activity (aw) and temperature (authors call this abiotic climate variables) for growth, sporulation, and OTA production in Aspergillus ochraceus group. Objective 2 of this study was to use existing published data against published data from similar strains under similar conditions. Differences observed in optimal conditions for growth, sporulation, and OTA production in two representative strains, A. westerdijkiae and A. steynii were obvious (Figure 1-3) and interesting. In addition, the authors suggest that the third order polynomial regression model can reliably predict growth (96%), sporulation (94.7%), and OTA levels (90.7%). In addition to the two experimental strains, the authors validated the models against 3 other strains from the A. ochraceus group and found that the model can predict outcomes between a regression coefficient (R2) range of 0.8819 to 0.9978. The authors conclude that effective and robust predictive models can be used to prevent or reduce mycotoxin contamination and to characterize a variety of abiotic factors beyond aw and temperature to also account for storage conditions, use of fungicides, genetic diversity and others. Outcomes from this work will help predict and eventually control OTA occurrence and contamination on important food and feed crops.
I have highlighted some strengths in this study as well as some minor comments and suggestions.
Strengths.
- Authors demonstrated clear goals and motivation to conduct study, title is clear and specific
- Strength in two-way interaction (water activity and temperature) on phenotypic characterization (growth, sporulation, OTA production), and use of two representative strains of ochraceus group, A. westerdijkiae 23971 and A. steynii 23096
- The authors did a nice job on the experimental design and use of experimental controls and replicates
Limitations (the authors should include this in discussion).
- Missing discussion on limitations of data interpretation, for example-
- The authors only used one standard in vitro laboratory treatment condition on YES solid medium (how might the effect of rich v. minimal medium affect model prediction, food-derived medium, food matrices)
- What about genotypic characterization (OTA gene expression)?
Minor comments throughout text.
Abstract.N/A
Introduction.
L48 …proven, would suggest that authors use ‘demonstrated or observed’
L51-54 Do authors know the mechanism to this observation re: differences in OTA levels between strains?
Paragraph 3 on mathematical models – should include what third order polynomial regression models are and what are the types of other models used to predict mycotoxin contamination in laboratory settings and food matrices.
Materials and Methods.
L82 – can authors include specifics what ‘similar in morphology’ means? Does this mean spore color? Microscopy? Spore pigmentation?
L83 – could authors briefly state and summarize how these two strains are different based on OTA cluster and phylogeny?
Comment. how genetically similar/different 23971 v. 23096?
Results and Discussion.
L254-6 Could also be due to genetic diversity and strain specificity
L260-2 In my understanding, growth and sporulation (development) genes are mechanistically connected to regulation of toxin production (at least in aflatoxin biosynthesis), “another group reported that there was no significant correlation between fungal growth and OTA production of A. westerdijkiae and A. steynii grown on food product based media”; could authors hypothesize what could be causing these differences in interpretation of how fungal development is associated (or not) with OTA production?
Figure 3 – could the increased levels of OTA observed in strain 23971 be interpreted as YES is a more conducive medium for toxin production for 23971 but not for 23096?
Comment. Based on data obtained in Fig. 1-3 and differences in two candidate strains tested, it appears that although strains have similar morphology (L82), differences observed as an effect of aw and temp on growth, sporulation, and OTA levels are driven by different in OTA cluster and phylogeny (genetic differences).
Minor comment. Third order; is this with – or without?
Author Response
We would like to thank you for valuable suggestion, we are really appreciated; here are responses to your comments:
Point 1:
Limitations (the authors should include this in discussion).
- Missing discussion on limitations of data interpretation, for example-
- The authors only used one standard in vitro laboratory treatment condition on YES solid medium (how might the effect of rich v. minimal medium affect model prediction, food-derived medium, food matrices)
- What about genotypic characterization (OTA gene expression)
Response 1:
Limitations have been added to discussion and highlited
Point 2:
L48 …proven, would suggest that authors use ‘demonstrated or observed’
Response 2:
The text has been modified.
Point 3:
L51-54 Do authors know the mechanism to this observation re: differences in OTA levels between strains?
Response 3:
The diversity among OTA-producing species might be explained by differences in their responses regarding growth and regulation of OTA biosynthesis to environmental conditions, biotic and abiotic.
Point 4:
Paragraph 3 on mathematical models – should include what third order polynomial regression models are and what are the types of other models used to predict mycotoxin contamination in laboratory settings and food matrices.
Response 4:
More details have been added to introduction
Point 5:
L82 – can authors include specifics what ‘similar in morphology’ means? Does this mean spore color? Microscopy? Spore pigmentation?
Response 5:
The text has been modified and highlighted
Point 6:
L83 – could authors briefly state and summarize how these two strains are different based on OTA cluster and phylogeny?
Response 6:
The text has been modified and highlighted
Point 7:
L254-6 Could also be due to genetic diversity and strain specificity
L260-2 In my understanding, growth and sporulation (development) genes are mechanistically connected to regulation of toxin production (at least in aflatoxin biosynthesis), “another group reported that there was no significant correlation between fungal growth and OTA production of A. westerdijkiae and A. steynii grown on food product based media”; could authors hypothesize what could be causing these differences in interpretation of how fungal development is associated (or not) with OTA production?
Response 7:
The text has been modified and highlighted
Point 8:
Minor comment. Third order; is this with – or without?
Response 8:
Third order with -. The text has been modified and highlighted through manuscript.
Reviewer 2 Report
I believe that the manuscript ID microorganisms-1207173 entitled “Predictive modelling and validation on growth, production of asexual spores and ochratoxin A of Aspergillus ochraceus group under abiotic climatic variables (temperature and water activity)” is a very interesting work because they provide a detailed analysis, it is well written with many experiments made by the authors who provide interesting data for the researchers. The results are valuable for the practice. Therefore, I consider that the manuscript is sustainable to be consideration for publication.
Manuscript microorganisms-1207173 has an interesting topic. The predictive models generated for the combined effect of temperature and water activity (aw) on growth, spore production, and OTA production of A. westerdijkiae and A. steynii and the validate of generated models against published data from other strains within the same genus in the literature can provide interesting data for the readers. Authors investigate fungal growth rate, sporulation, and OTA production as a function of aw and temperature and based on data generate predictive models that are then validated.
The manuscript is generally well written, with a logic structure and the novelty of the study is clearly defined. The methods applied in the study are suitable to investigate the combined effect of temperature and water activity (aw) on growth, spore production, and OTA production and to generate and validate predictive models. The description of materials and methods are clear. The experimental results are significant and interesting being discussed in detail with relevant references. The results are valuable for the practice.
Author Response
Thanks for your comment, we are really appreciated
Reviewer 3 Report
The manuscript presented for review is very chaotic and careless. It would have to be rewritten. Title is too long and becomes illegible. The abstract does not bring us closer to the research problem, it is written carelessly. Keywords must be improved, words from the title should not be repeated. The introduction does not provide significant information about the toxin, what is the big problem on a global scale? Is it possible to limit these fungi. In the methodology, apart from the information that each variant was repeated three times, there is no precise information on how many samples were used for the analysis. Was it only 3 variants in 3 repetitions? For me, the model of the experiment is not clear. Please order the record of the units. My knowledge of mathematical models is not sufficient to assess this aspect at work. Unfortunately, although the results may interest readers, the presentation is discouraging.
Author Response
Thanks for your comments; here are responses to your comments:
Point 1:
The manuscript presented for review is very chaotic and careless. It would have to be rewritten.
Response 1:
Thanks for your critical comment. We regret to disagree the reviewer comment that the manuscript is not well organized. The manuscript has been rigorously reviewed, structured and the language was checked by native English speakers.
Point 2:
Title is too long and becomes illegible.
Response 2:
The title has been shortened and highlighted.
Point 3:
The abstract does not bring us closer to the research problem, it is written carelessly.
Response 3:
Our abstract summarizes the overall purpose of the study, investigates the basic design of the study, major findings and conclusion.
Point 4:
Keywords must be improved, words from the title should not be repeated.
Response 4:
Some of the keywords have been changed.
Point 5:
The introduction does not provide significant information about the toxin, what is the big problem on a global scale? Is it possible to limit these fungi.
Response 5:
The introduction provides information about
- What is OTA and its impact on health
- Ochratoxin A producers and which food matrices can contaminate
- Why there two species ( westerdijkiae and A. steynii) are important for the study
- Responses of OTA producers to abiotic climatic factors and their interactions
- Paragraph on mathematical models
Point 6:
In the methodology, apart from the information that each variant was repeated three times, there is no precise information on how many samples were used for the analysis. Was it only 3 variants in 3 repetitions? For me, the model of the experiment is not clear. Please order the record of the units.
Response 6:
The materials and methods have been clearly described.
The data were divided into three sets. The data produced from experimental work of Aspergillus westerdijkiae IBT 23971 were used for the training set, the data produced from experimental work of Aspergillus steynii IBT 23096 were used for the testing and derived data of Aspergillus ochraceus IBT 11952, Aspergillus steynii IBT 22339, and Aspergillus west-erdijkiae IBT 10738
Point 7:
My knowledge of mathematical models is not sufficient to assess this aspect at work. Unfortunately, although the results may interest readers, the presentation is discouraging.
Response 7:
The experimental results have been presented in a logical sequence and being discussed in detail with relevant references.
Round 2
Reviewer 3 Report
Thank you for the submitted manuscript. In my opinion it is much better than the original version, a revised version can be published